

# Detection of Berezinskii-Kosterlitz-Thouless transition
# via generative adversarial networks

**Daniele Contessi[1,2]⋆, Elisa Ricci[3], Alessio Recati[1] and Matteo Rizzi[2]**

**1** Dipartimento di Fisica, Università di Trento & INO-CNR BEC Center, 38123 Povo, Italy
**2** Forschungszentrum Jülich GmbH, Institute of Quantum Control,
Peter Grünberg Institut (PGI-8), 52425 Jülich, Germany
Institute for Theoretical Physics, University of Cologne, D-50937 Köln, Germany
**3** Dipartimento di Ingegneria e Scienza dell'Informazione, Università di Trento, & Deep
Visual Learning research group, Fondazione Bruno Kessler (FBK), 38123 Povo, Italy

⋆ daniele.contessi@unitn.it

## Abstract

The detection of phase transitions in quantum many-body systems with lowest possible prior knowledge of their details is among the most rousing goals of the flourishing application of machine-learning techniques to physical questions. Here, we train a Generative Adversarial Network (GAN) with the Entanglement Spectrum of a system bipartition, as extracted by means of Matrix Product States ansätze. We are able to identify gapless-to-gapped phase transitions in different one-dimensional models by looking at the machine inability to reconstruct outsider data with respect to the training set. We foresee that GAN-based methods will become instrumental in anomaly detection schemes applied to the determination of phase-diagrams.



# 1 Introduction

Recently we have experienced an explosion in the application of machine learning related techniques to face physical problems and in particular in quantum many-body and statistical physics [1, 2]. The relevance of these techniques is grounded in the ability of the machines to process extremely complex data in an efficient way for the sake of pursuing classification tasks, pattern recognition and even generation of brand new data responding to some constraints or development of decision processes. In spite of their impressive achievements, the research is still in constant and hectic evolution and a great benefit is represented by the application of these techniques to fields beforehand unexplored. Physics is an extremely fruitful discipline in this sense. The common aim of manipulating and studying complex data allows a sharing of expertise that leads on the one hand to reveal new physics and enhance the power of analysis and on the other hand it helps the machine learning community improve the techniques and understand the intriguing mechanisms of artificial learning.

One of the specific tasks in physics that is undergoing to a huge effort in machine learning application is the phases recognition and classification. There is plenty of very recent works concerning the study of phase transitions both with supervised and unsupervised algorithms [3–12]. Indeed, the exploration of phase diagrams has always been a relevant physical problem both from a numerical/analytical and experimental point of view. The detection of a phase transition of a system normally consists in the measure of specific quantities which reflect some changes in the physics through the transition from one thermodynamic state to another. We believe that the most relevant strengths of machine learning algorithms in generalization and discovering of hidden patterns can perfectly suit the scope. In particular, protocols like anomaly detection have lately demonstrated to be a powerful tool for the mapping of physical systems' phase diagrams in a completely unsupervised manner and with very little or even no prior physical information about the kind of phases under consideration.

In this work, we develop a new architecture able to map the phase diagram of physical systems through an anomaly detection scheme. Our method is based on Generative Adversarial Networks (GANs) and improves the networks adopted by the previous works of [10, 13] because of the different training procedure that inherits the improved modern standards from the machine learning community. We test our method with the detection of the Berezinskii-Kosterlitz-Thouless (BKT) phase transition in three different quantum systems at zero temperature. The very peculiar aspects of such a transition have been object of intense theoretical and experimental studies since its first prediction for the classical two-dimensional XY model [14, 15]. The absence of an explicit symmetry breaking makes its characterization beyond the Landau-Ginzburg criterion. Moreover, its gapless-to-gapped nature implies a very slow convergence with the system's size and a smooth behaviour of the thermodynamic quantities. For these reasons, the determination of the critical parameters has always been a challenging task if compared with other kinds of phase transitions. Recently, some explicit attempts to address this task by means of machine learning algorithms were made [10, 11, 16–20].

The choice of the input features for the network is crucially linked with the success of the

learning outcome together with the skeleton of the architecture. We choose to employ the entanglement spectrum (to be defined below), a very general quantity related to the degree of quantum correlations among the sub-portions of the system. Its use was originally proposed in [5] as input data on which the unsupervised detection of patterns leads to the identification of phase transitions. The study of entanglement properties had indeed a huge reception in the physics community for the characterization of quantum many-body systems and has driven the escalation of the quantum information theories. We provide a detailed justification of this choice as well as some physical insights on what is happening to the features the network is fed with during the transition.

Our approach is able to detect the BKT transitions in the XXZ spin chain, the Bose Hubbard model and its generalization for the two species case. Despite the microscopic differences, such models share a phase whose low-energy physics is described by Tomanaga-Luttinger liquids. Following the intuition of [21] we know that when such a description is valid, the entanglement spectrum exhibits very peculiar properties that are lost once the system undergoes the BKT phase transition. Therefore we train the GAN architecture by passing the (lower part of) the entanglement spectrum, and show that it is able to properly signal the emergence of anomalies in the spectrum when one crosses the BKT critical point and enters the gapped phases. It is worth stressing that such a task is performed without any *a priori* knowledge (order parameter, relevant correlation functions...) about the phases which break the Tomonaga-Luttinger liquid description. All the physical information is contained in the entanglement spectrum without further manipulation.

The method is not thought to compete with the most traditional methods for the detection of BKT transitions concerning the numerical precision. It is rather a way to identify with little prior knowledge the presence of different phases – as we show – even for small system sizes, and for subtle transition as the here studied BKT.

## 2   Entanglement Spectrum as dataset

In this work we deal with different quantum many-body systems at zero temperature. Because of the huge number of degrees of freedom, their characterization always requires a selection or a compression of the information in order to study their properties. We want to take into account a universal quantity that does not need the combination with any additional knowledge of the concrete system. For this purpose, we choose the degree of quantum correlation – or entanglement – among the internal components of the many-body systems that has been proven to contain a wealth of physical information.

There exist many measures of quantum entanglement and a big subset of the possibilities is based on the quantification of the correlations among sub-portions of the system. While still lots of questions are open concerning the multipartite case, the situation is now very clear for the bipartite entanglement of pure states. Specifically, the system is divided in two complementary sub-systems A (the new system) and B (the environment) and a measure for the degree of entanglement between them can be introduced. The most famous one is the Von Neumann entropy, that is exactly related to the Shannon entropy in information theory. It can be computed as $S_{VN} = -\text{Tr}_A \rho_A \log \rho_A$ ($\rho_A \equiv \text{Tr}_B \rho$ being the reduced density matrix for the A subsystem). For the properties of the trace, a simpler expression for the entropy is obtained via diagonalization of the reduced density matrix $\rho_A$ in terms of its eigenvalues $\lambda_i$ which are probability values. The flatness or steepness of their distribution is accounted for in the Von Neumann entropy for the sake of measuring the degree of entanglement. For example, a completely flat distribution witnesses a maximally entangled state, while a fully peaked one is associated to a product state, i.e. the quantum correlations between the two partitions are

respectively maximum and minimum. Their distribution can be interpreted as a Boltzmann distribution of an effective entanglement Hamiltonian, in other words the transformed eigenvalues $\xi_i = -\log(\lambda_i)$ constitute the energy spectrum of the entanglement Hamiltonian that takes the name of entanglement spectrum (ES) [22, 23]. From now on, for simplicity we will refer both to the entanglement Hamiltonian eigenvalues $\xi_i$ and the reduced density matrix eigenvalues $\lambda_i$ as ES since they are related through a map.

Besides the trivial cases, it has been highlighted in several contexts that the ES exhibits relations and degeneracies which reflect very peculiar aspects of the physics and the symmetries of the system [24]. For this reason, the study of its structures already revealed itself as a powerful tool for the detection of changes in the physics of a system along a quantum critical point. The advantages are manifold: the ES is not directly related to any local order parameter nor correlator and it is not necessary to know the topological character of the phases – the topological order actually emerges automatically from the ES – in order to deal with it. The absence of prior knowledge on the system makes this choice very general and model-independent for all the cases in which the access to the ES is available, in contrast to the more traditional methods for the phases detection. In this work, we want to bring the study of the ES structures to another level: their interpretation is not human but machine driven. In particular, we want to perform the detection of a phase transition by means of machine learning algorithms in an unsupervised fashion. We develop a GAN architecture that takes as input feature the ES and is able to detect the changes in its structures when the system undergoes a phase transition.

## 3 GAN for unsupervised anomaly detection

The idea behind the method is based on the same principles of Refs. [10, 13]: the detection of the phase transition relies on the ability of reconstruction of the ES. The principal ingredient for this purpose is the deep neural network auto encoder (AE) which maps every vector of real positive eigenvalues of the ES included in the $[0, 1]$ interval (that we call $x$) to another vector ($z$) in a low-dimensional latent space through a parametrized function $f_\theta$ and maps back $z$ to the original space with another parametrized function $g_\phi$:

$$\hat{x} = g_\phi(f_\theta(x)) = g_\phi(z). \tag{1}$$

The parameters $\phi$ and $\theta$ are optimized during the training procedure on samples that belong to a chosen region of the phase diagram (normal samples) meaning that the AE learns to reconstruct the original input in that region. The ability of reproducing the normal samples is lost when the AE faces an abnormal vector. In order to quantify this ability, we look at the reconstruction (or reproduction) loss between the input $x = (\lambda_1, ..., \lambda_N)$ and the output $\hat{x} = (\hat{\lambda}_1, ..., \hat{\lambda}_N)$:

$$\mathcal{L}_{rec}(x, \hat{x}) = \sqrt{\sum_j (\lambda_j - \hat{\lambda}_j)^2}. \tag{2}$$

In this way we choose the *anomaly score* that indicates the presence of an abnormality in a sample if it is greater than a certain threshold at test-time. In other words, if the AE is correctly trained, it has learnt a probability distribution from which the training examples are sampled and can detect when an example does not belong to the same distribution by quantifying its being non-conforming to it. In the exploration of a phase diagram, this protocol has several advantages because, for instance, it allows to choose the region of the normal data where their production is numerically favourable or where the physics of the system is well understood. Nevertheless, it allows for a scan of the phase diagram in an unsupervised way because no prior knowledge of the possible phase labelling is required.

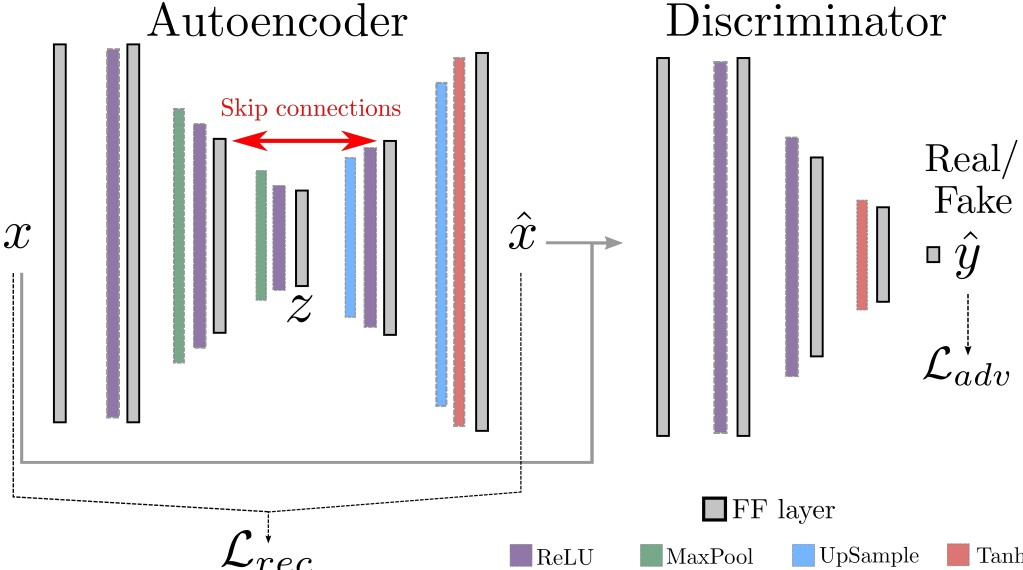

Figure 1: Sketch of the GAN architecture we used for the phase transition detection. The AE takes in input a vector with the eigenvalues of the ES ($x$) and operates an encoding and a decoding to its reconstructed version $\hat{x}$. The discriminator takes in input $x$ or $\hat{x}$ and gives a real number $\hat{y}$.

The usual training procedure involves the minimization of the reconstruction loss but it has been demonstrated that the adversarial training usually improves the optimization of the parameters and entails a better reconstruction. We decide to provide the AE with a *discriminator* that takes $x$ or $\hat{x}$ as input and outputs a single real number $\hat{y}$. The composite structure is a GAN [25] and is depicted in Fig. 1. The role of the discriminator is to distinguish if the vector in input is a real example (one ES from the training set) or a fake one (a reconstructed ES from the output of the AE). For real examples the output of the discriminator should reach $\hat{y} = 1$ whereas for fake examples $\hat{y} = 0$. The two networks are trained together in a competitive way following this scheme for every batch of training samples:

- the discriminator is optimized alone in order to be able to predict that $N_{batch}/2$ samples from the dataset are real and that their reconstructions by the AE are fake

- the AE – that is a full-fledged *generator* of samples – is optimized in order to fool the discriminator, hence making it predict that the remaining $N_{batch}/2$ reconstructed samples are real.

During the training, the following adversarial loss is minimized, jointly with the the reconstruction loss:

$$\mathcal{L}_{adv}(y, \hat{y}) = -[y \log(\hat{y}) + (1 - y) \log(1 - \hat{y})], \tag{3}$$

where $y$ is the target value of the discriminator. As in common practise for actualizing the adversarial competition, we separately train the sub-networks by freezing the respective parameters' optimization. Since in general the reconstruction loss and the adversarial loss range within intervals of different magnitudes, we combine them adding some weights in the overall loss:

$$\mathcal{L}_{tot} = \lambda \mathcal{L}_{adv}(y, \hat{y}) + \epsilon \mathcal{L}_{rec}(x, \hat{x}). \tag{4}$$

The hyperparameters $\lambda$ and $\epsilon$ are tuned in order to adjust the influence of the individual losses as well as the learning rates of the two subnetworks' optimizers. At test-time the discriminator is not necessary anymore: the AE – that have benefited from the adversarial training – faces

the whole phase diagram and the anomaly score is computed. We employ a GAN architecture implemented with PyTorch [26][1]. Among all the possible structures for the layers, we tried to adopt a Convolutional Neural Network (CNN) whose principle is to convolve the layers with some filters but we find that the Dense structure over-performs the CNN for our task. The reason must probably be ascribed to the full-connectivity meaning that every neuron of each layer is connected with all the neurons of the previous and the following ones. In the AE, the MaxPool filter is added for down-sampling the features in the encoding process (by considering the biggest values and neglecting the smallest ones in a fixed size sliding window) and for the decoding part the UpSample is used for going back to the original space (it fills the holes created by the up-sampling to a bigger dimension through customized interpolations). Moreover, a skip connection between two latent layers is added to improve the model performance. It consists in the transfer of an encoding layer's information, i.e. for example $x_E$, to the same-size layer in the decoder $x_D$ before the activation function. In Fig. 1 this is represented with a red arrow. Skip connections were first introduced to prevent gradient vanishing in deep networks and then in residual networks in order to overcome the so-called degradation problem (see [27, 28] and references therein). Supposing that $x_E$ is transformed toward the decoder with a function $F(x_E)$ and the symmetric layer in the decoder $x_D = \sigma(F(x_E))$ is obtained by applying an additional non-linearity $\sigma$, the skip connection:

$$x_D = \sigma(F(x_E) + x_E),$$

does not introduce neither extra-parameters nor computational complexity. Nevertheless, the optimization of the new $F$ (i.e. the *residual* mapping) is easier thanks to the information of $x_E$ and, more specifically, was found to be responsible of the learning of good abstract representations in the bottleneck, rather than local details of the data [29]. All the manipulations on the data are made through standard built-in methods of the library.

To set out even more clearly and in mathematical terms, the $f_\theta$ function of Eq. (1) can be written in its expanded form combining different fully connected modules of the form $h(W_\gamma \cdot x + b_\gamma)$ where $W\gamma$ is a parametric weight matrix, $b_\gamma$ is a bias vector and $h$ is a non-linear activation function. So $z = f_\theta(x)$ is written as:

$$z = \text{ReLu}\left\{W_\theta^{(3)} \cdot \text{MaxPool}\left[\text{ReLu}\left(W_\theta^{(2)} \cdot \text{MaxPool}\left[\text{ReLu}\left(W_\theta^{(1)}x + b_\theta^{(1)}\right)\right] + b_\theta^{(2)}\right)\right] + b_\theta^{(3)}\right\}, \quad (5)$$

with $i$ the index of the layers. The $h$ non-linear functions used are ReLu and Tanh as indicated in Fig. 1.

To our knowledge, this is the first time that an adversarial training of the AE is implemented via a GAN architecture in order to address the problem of the phase transition detection.

## 4 Description of the models

We are now set to discuss the application of the techniques to the BKT phase transition. Its elusive and usually numerically demanding nature derives from the exponentially slow gap opening and the smooth behaviour of the thermodynamic quantities, which call for quite sophisticate scaling analysis and often for calculations with twisted boundary conditions [30].

We decide to apply our method to three different one-dimensional (1D) quantum systems on a lattice at zero temperature. The bipartite entanglement properties for 1D systems that are described by a 1+1 conformal field theory (CFT) are now well known [31]: on one side it has been highlighted that the entanglement entropy scaling with the size of the subsystem

---

[1]The code is available together with the dataset at https://github.com/cerbero94/GAN_CP.

depends on the central charge [32]; on the other side, in different contexts it has been demonstrated that the structures in the full ES contains even more: it can be used as a fingerprint of topological order [22, 23], the difference between the two largest non trivially degenerated eigenvalues (Schmidt gap) changes along a quantum critical point [33, 34] and the lower part of real-space ES is organized in interesting CFT structures [21] (see also Fig.2). In light of the fact that the transition we want to observe occurs between a Tomonaga-Luttinger liquid (described by a CFT as above) [35] and a gapped state, we know from [21] that the mentioned ES structures should disappear after the transition. This fact fits with the idea of using an anomaly detection protocol to locate the phase transition: the machine learns the structures of the ES in one phase and when it faces anomalies in these structures – i.e., when the physics of the system changes – it reports that there has been a deviation from the training dataset's normal behavior. The variation of these features normally manifests clearly in the thermodynamic limit. Even though the small changes in the ES for the accessible sizes of the physical systems would make the discernment between two phases a delicate task for the human eye, the machine succeeds in carrying out the distinction.

### 4.1 XXZ model

The first model we investigate is the XXZ model, which is exactly solvable. The boundaries of the phase diagram are therefore precisely known (for very good reviews see [36, 37]) and it can be used as a benchmark. The Hamiltonian for the spin $1/2$ chain of $L$ sites is:

$$H_{XXZ} = -J \sum_{j=1}^{L-1} \left[ \frac{1}{2}(S_{j+1}^+ S_j^- + S_j^+ S_{j+1}^-) + \Delta S_{j+1}^z S_j^z \right], \tag{6}$$

where $S^\alpha = \frac{1}{2}\sigma^\alpha$ ($\sigma^\alpha$ being the Pauli matrices). By changing the magnitude of the anysotropy $\Delta$, the model has a ferromagnetic ground state ($\Delta > 1$), a paramagnetic behaviour that is described by a Luttinger liquid ($-1 < \Delta < 1$) and an anti-ferromagnetic ground state for $\Delta < -1$. We are interested in the transition between the Luttinger liquid phase and the anti-ferromagnetic one because it belongs to the BKT universality class. Instead, the transition from the paramagnet to the ferromagnet is first-order and therefore much less difficult to identify.

### 4.2 Bose Hubbard model

The second model we study is the Bose-Hubbard model (BH). The phase diagram of this non-integrable model has been thoroughly analized in literature and, hence, it represents a good benchmark for our unsupervised phase recognition method. Detailed information on both the numerical and experimental aspects about this method can be found in reviews [38, 39]. The Hamiltonian for a $L$-sites system is:

$$H_{BH} = -J \sum_{j=1}^{L-1} (b_{j+1}^\dagger b_j + \text{h.c.}) + \frac{U}{2} \sum_{j=1}^{L-1} n_j(n_j - 1), \tag{7}$$

with $b_j^\dagger$ ($b_j$) the bosonic creation (annihilation) operator at site $j$ and $n_j = b_j^\dagger b_j$ the number operator. For fixed unitary filling $\nu = \langle n_j \rangle = 1$, the ground state behaves like a 1D superfluid below $U/J \simeq 3.39$ [40] whose low-energy spectrum is effectively described by a Luttinger liquid. The transition from the superfluid phase to the Mott insulator phase at this filling is a BKT transition too.

## 4.3 Two-component Bose Hubbard model

The last model we address is the generalization of the BH for two-species (A and B), i.e., two Bose-Hubbard models coupled via a local density-density interaction term. The Hamiltonian reads:

$$H = \sum_{\alpha=A,B} \sum_{j=1}^{L} \left[ -\left( t_\alpha b_{j+1,\alpha}^\dagger b_{j,\alpha} + \text{h.c.} \right) + \frac{U_\alpha}{2} n_{j,\alpha}(n_{j,\alpha}-1) \right] + U_{AB} \sum_{j=1}^{L} n_{j,A} n_{j,B} \,. \tag{8}$$

The phase diagram of the two-species Bose Hubbard model (BH2S) is very rich and it includes pair-superfluidity, counterflow superfluidity, phase separation, charge density wave quasi order and supersolidity [41, 42], whose boundaries in are still not well determined. We focus on a small region of the phase space by choosing a $\mathbb{Z}_2$ symmetric mixture with hopping parameters $t_\alpha = 1$, fixed intra-species on-site repulsions $U_\alpha = U = 10$ and total unitary filling $\nu_A = \nu_B = 0.5$. For $U_{AB} < 0$ the ground state is either composed of two superfluids (2SF) or of a single superfluid of couples – namely pair-superfluidity (PSF) – for inter-species attractions $U_{AB}$ small enough to prevent collapse [43]. In the 2SF phase, the low energy spectrum is described by two Luttinger liquids corresponding to gapless modes in the density (in-phase) and spin (out-of-phase) channels. Through the transition to PSF, the spin channel acquires a gap and undergoes to a BKT phase transition whereas the density channel remains gapless. It is worth to mention that the position of the quantum critical point is not only still unknown but it is not completely clear [44] if the transition occurs just below $U_{AB} < 0$ or for a finite value of $U_{AB}$ in the thermodynamic limit. In this work we provide numerical evidence that is occurs for a finite $U_{AB}$ for every size we simulate, in support to out previous work [43] with a completely different method.

# 5 Results

We deal with the many-body problem for finite sizes of the systems through a Matrix Product States (MPS) ansatz. It is always possible to extract the eigenvalues of the reduced density matrix $\lambda_i$ from the tensor network by means of contractions of the tensors and singular value decompositions (SVDs) without altering the physical content of the stored information [45, 46]. This is possible for every physical bipartition and specifically we choose a symmetric bipartition that cuts the system in the middle.

## 5.1 Features of the dataset

We show in Fig. 2 the ES for the BH and the BH2S in the Luttinger liquid regime. We display the logarithm with base 10 of the reduced density matrix eigenvalues $\xi = -\log_{10}(\lambda_i)$, which are related to the eigenvalues of the entanglement Hamiltonian, as function of the particle number sector (for the XXZ model being the number of spinless interacting fermions resulting from the mapping of the Hamiltonian under Jordan-Wigner transformations [37]). The choice of the base 10 is just related to the more convenience in the order of magnitude readout. Since there are multiple eigenvalues for each sector, they can be ordered in a decreasing way: we label every eigenvalue with the (shifted) sector number, $\delta N = N - L$, and with the sorting index, $k$.

For the XXZ and the BH model, the structures of the ES at fixed $k$ exhibit a parabolic envelope and the parabolas with different $k$s have the same curvature. Besides, the parabolas are translated by multiples of the difference between the first and the second eigenvalues of the $\delta N = 0$ sector:

$$\Delta \xi_0 = \xi_{\delta N=0}^{k=1} - \xi_{\delta N=0}^{k=0} \,. \tag{9}$$

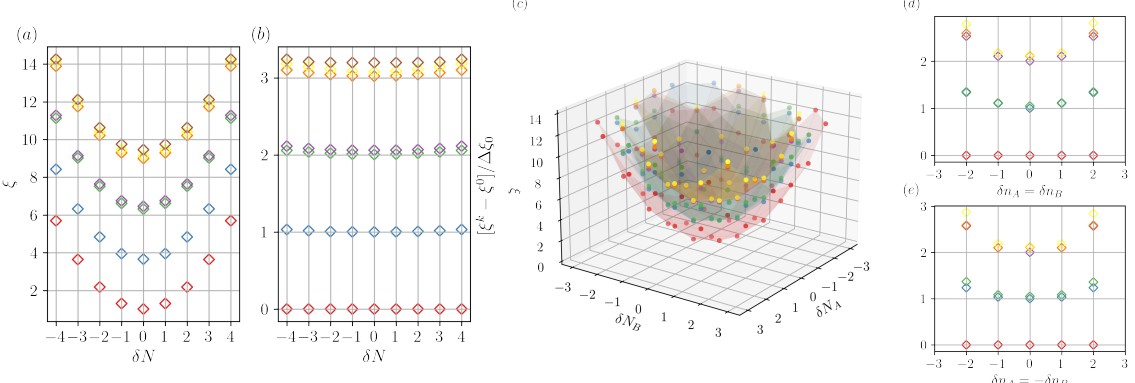

Figure 2: Entanglement spectrum for the $L = 256$ BH chain with open boundary conditions at $U/J = 1$ ((a) and (b)) and for the $L = 96$ BH2S chain with periodic boundary conditions at $U_{AB}/U = -0.1$ ((c), (d) and (e)). In (a) the parabolic envelopes' structure is the marker of the emergent CFT. For the BH2S model the parabolas become parabolic surfaces on the $\delta N_A/\delta N_B$ plane (in (c)). Panel (b) shows the same data of (a) obtained by subtraction of the lowest parabola $\xi^{k=0}$ and by setting the distance between $\xi^{k=1}_{\delta N=0}$ and $\xi^{k=0}_{\delta N=0}$ equal to one in order to better appreciate the equally spaced structure and the degeneracies. In (d) and (e) the equivalent procedure as in (b) is applied to the diagonal sections of the paraboloids which correspond to the density ($\delta N_A = \delta N_B$) and the spin channel ($\delta N_A = -\delta N_B$) respectively. After the phase transition to PSF, only the eigenvalues of the density channel (d) maintain the conformal behaviour while the spin channel acquires a gap as shown below in Fig. 3.

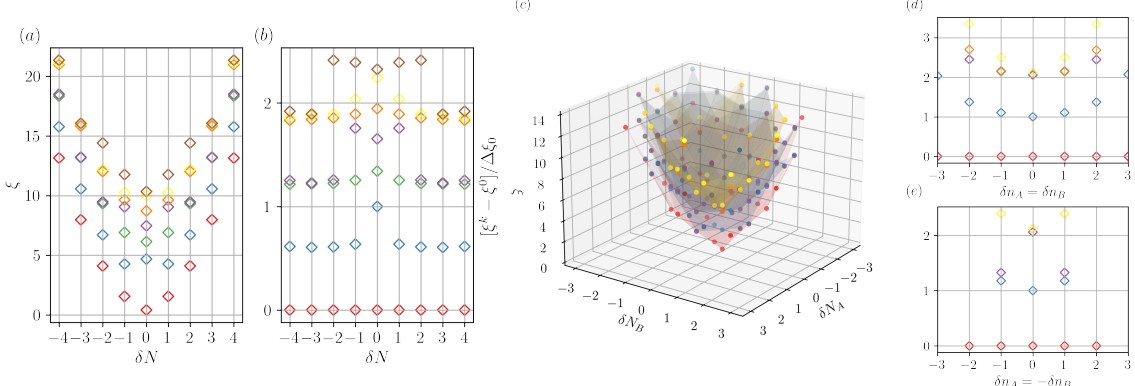

Figure 3: Entanglement spectrum for the same systems as in Fig. 2 but in the Mott insulator phase at $U/J = 4$ ((a) and (b)) and in the PSF phase at $U_{AB}/U = -0.35$ ((c), (d) and (e)). In (a), (b) and (e) the behaviour of the ES is clearly no more parabolic.

As a consequence, by subtracting every parabola by the first one ($k = 0$) and by rescaling with $\Delta\xi_0$, the manipulated ES displays some equally spaced structures with integer distances whose degeneracies are explained through a parallelism with energy spectra of boundary CFTs [21]. These conformal structures are then destroyed after the phase transition as can be seen by comparing Fig. 2 and Fig. 3.

Despite the apparent ease of description, the extraction of such parabolas (and their degeneracy counting) from a finite size system is far from being an easy task, since the convergence to thermodynamic limit is, once more, extremely slow. For this reason, the ability of generalization typical of a machine driven approach is instrumental in carrying out the pattern recognition for pursuing the distinction of the different regimes.

More interesting properties appear in the generalization to the BH2S model where the situation is slightly more complicated. The symmetries associated to the A and B number of particles' conservations provide two labels for the sectors $\delta N_A$ and $\delta N_B$. The parabolas become equally spaced paraboloids in the region where the two Luttinger liquids coexist. As already anticipated, once the system undergoes the phase transition to PSF the gap opens only in the spin channel. Indeed, as clear from Fig. 3 after the opening of the gap, the arrangement of the ES eigenvalues on the density channel slice of the paraboloids ($\delta N_A = \delta N_B$) still behaves according to the above-mentioned conformal structures while it loses these fascinating properties in the spin channel ($\delta N_A = -\delta N_B$). It is furthermore relevant that in the PSF the parabolic envelopes observed in the density channel are the only preserved while the ES structures do not exhibit parabolas for any other direction on the $\delta N_A/\delta N_B$ plane. We did not find any analysis about this generalization in the literature.

## 5.2 BKT detection

In order to carry out the BKT detection, we consider the biggest eigenvalues of the ES in magnitude for various system's sizes for all the three different models. Their sorting is done in a decreasing way for the first ES corresponding to the origin of the phase diagram. Then all the spectra are sorted along the same sequence of symmetry sectors, without however passing the charge value to the GAN. Therefore it has no knowledge about the underlying parabolic structures and the whole procedure cannot be ascribed to a simple fitting of them. We divide a portion of the data in the Luttinger liquid regime in two disjoint datasets: training set and validation set. The training was always performed with samples belonging to the regions where the system behaves like Luttinger liquids, therefore where the input features show the above-described peculiar properties. The validation points are instead taken from an interval of the parameter space after the training toward the transition. With this splitting, we set a threshold both for the training loss ($\bar{\mathcal{L}}_{rec}^{tr}$) and the validation loss ($\bar{\mathcal{L}}_{rec}^{val}$) at training time (see the App. A for numerical details). The idea behind this procedure is that if we pick the validation points sufficiently far apart from the phase transition, their loss will remain low even if the network does not account for them for the optimization. In this way the training and validation data are always coherent between each other for different system's sizes with respect to a random reshuffle and the criterion of setting a threshold makes sense. Otherwise it could happen that for some special random splitting the losses are particularly low or high depending on their arrangement on the phase space. Moreover, by looking for a low loss in the validation set we check that the network reaches an optimization good enough to generalize well just outside the training dataset. For the XXZ, the training samples for each system's size are taken from the region $-0.65 \leq \Delta/J \leq 0$ (about 550 points) and the validation ones from $-0.8 \leq \Delta/J < -0.65$. The BH training consists in about 500 points from the interval $0 \leq U/J \leq 2.5$ and the validation from $2.5 < U/J \leq 3$ while for the BH2S model there are about 200 samples from the region $-0.1 \leq U_{AB}/U \leq 0$ as training points and the validation ones are from $-0.15 \leq U_{AB}/U < -0.1$. In order to check the stability of the algorithm, we verified that the results do not show substantial differences when the intervals are rescaled (see App. A for the details). Moreover, we found that the results are similar even if only one AE per model is trained on a single system's size and then faces the test sets for the other sizes. This procedure is applicable and could be useful when the computational production of the training dataset is particularly costly and a rough estimation of the phase diagram boundaries

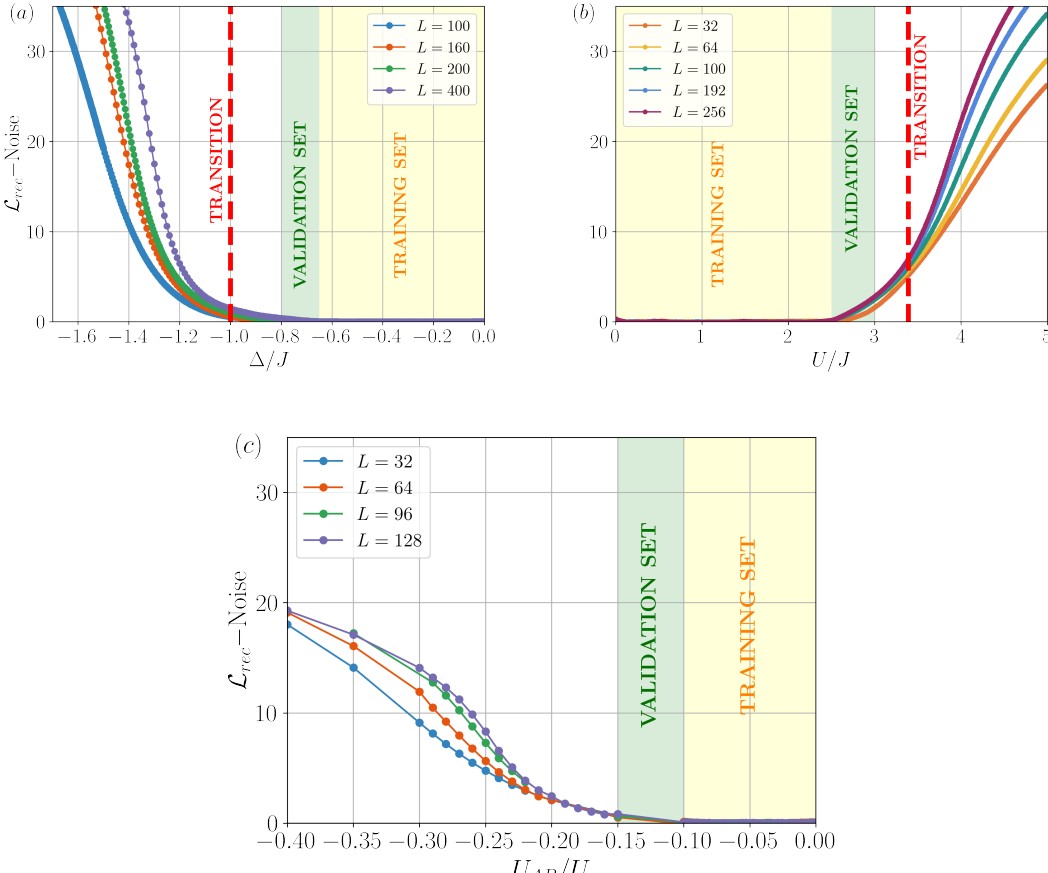

Figure 4: The reconstruction losses percentage purged of the noise as a function of the free parameters for the three models and for different sizes of the systems (indicated in the legends). The available values for the BKT transitions are reported with a dashed red line. The training sets include points in the yellow intervals whereas the validation samples are taken from the green intervals. Datasets of training samples are simulated for all the shown sizes. In $(a)$ the XXZ model: all the curves show an abrupt rise in the proximity of the transition in the thermodynamic limit. In $(b)$ we plot the result for the BH model. In this case the loss starts to rise after the training points but remains under the 10%, instead in the region of the transition the behaviour of the curves changes according to the size of the system. Even by changing the training window, the splitting of the curves remains very closed to the transition. In $(c)$ the BH2S model for which the predicted value for the transition is still uncertain. The plot shows a trend very similar to the single species BH model. We hypothesize that the transition should happen in the $-0.25 < U_{AB}/U < -0.2$ interval where the curves begin to separate in analogy to the previous case. This result is in good agreement with the prediction in [43].

is enough in a first approach.

The final reconstruction loss curve for each size is finally obtained by subtracting the loss computed on the full dataset by the trained network with the average training loss in order to purge the curves of the noise. The results are represented in Fig. 4. We have proven that the predictions and the final AEs really take advantage of the discriminator if compared to AEs that are not trained via adversarial competition. We are able to achieve training losses of the

$10^{-3}$ order within 250 epochs. It is fundamental to reach good values of precision for pursuing an accurate reconstruction of the ES's structures since the magnitude of the eigenvalues drops rapidly after the first leading ones. This level of precision is sufficient to faithfully reproduce at least the first two parabolas of the ES in the Luttinger regimes whereas the AEs become unable to reconstruct the structures after the phase transition.

All the results show that the loss experiences a rise in the region compatible with the phase transition. Besides, the increase depends on the size of the system and exhibits greater and greater steep going towards the thermodynamic limit. As a difference with respect to the previous works involving the same scheme [10, 13] in which the authors exploit tensor network techniques for the simulation of the thermodynamic limit, we cannot expect a sudden change in the ES structures both because of the finite size and the BKT nature of the transition. Anyway, by looking at their results for the first and second order phase transitions we suspect that the loss increase in those circumstances would be much more sharp. In any case, it is worth to point out that this method allows for an automate scan of the phase diagram that cannot be a substitute to usual numerical methods for the location of the transition (for the BKT see for example [40, 47]), therefore getting a comparable precise value for the critical parameters is beyond the scope.

## 6 Conclusion

In this work we show that the entanglement spectrum represents a reliable quantity to look at in order to perform a machine driven detection of the elusive BKT transition for one-dimensional quantum systems by injecting almost no prior knowledge about them. We face such a task by using a GAN that benefits from the adversarial training in an anomaly detection scheme. We find that such architecture is a valid candidate for the purpose of discerning the change in the physics associated to the BKT transition and capable to remarkably improve the performances with respect to the previously adopted networks.

With this work we make a step forward in the exchange of technology between the physics and the machine learning communities. A step that gives rise to new interesting applications that we leave for a future investigation, e.g. the analysis of the features and correlations that the machine recognises as relevant for the detection of a phase transition; the tackling of the same problem by using different input features for the machine; a study on how far the machine can go in the understanding of the phase transition nature and universality in order to guide the entrance of more sophisticated quantities to solve the problem (like e.g. the stiffness specifically for our case); the implementation of an hopefully more precise procedure for the location of the transition by combining the losses obtained from the training in all the different phases separately. While writing this manuscript we became aware that even better performances could be achieved by taking inspiration from the most modern techniques adopted by the machine learning experts such as GANomaly architecture, AnoGAN, etc. [48–50]. Let us conclude by remarking once again, that this approach is not supposed to become a substitute to the traditional methods for the phase transitions detection but allows to obtain a qualitative map of a phase diagram with very little knowledge about the nature and arrangement of the phases in an automatic fashion.

*Note added:* during the submission process of this manuscript we found out [51] that provides a slightly different approach to the detection of phase transitions in spin systems using a GAN architecture.

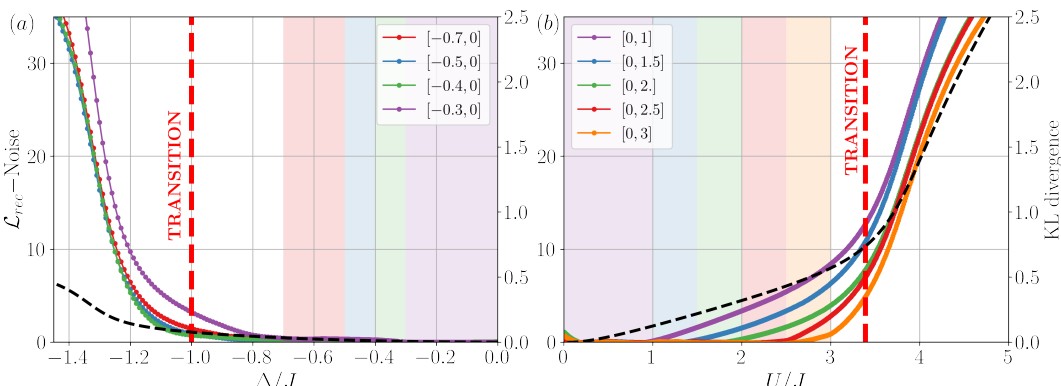

Figure 5: The loss curves for the XXZ with $L = 400$ sites in $(a)$ and for the BH model with $L = 256$ sites in $(b)$ for different training regions indicated in the legend and accordingly to the curves' color by the shaded regions. The KL divergence is represented with the black dashed line, whose values must be consulted in the rightmost y-axis.

## A  Numerical details and training windows

The training was performed by setting $\lambda = 0.1$ and $\epsilon = 10$ in Eq. (4). The typical threshold for the losses in the training and validation region for the XXZ model are $\bar{\mathcal{L}}_{rec}^{tr} = 0.00005$ and $\bar{\mathcal{L}}_{rec}^{val} = 0.0001$, for the BH model are $\bar{\mathcal{L}}_{rec}^{tr} = 0.005$ and $\bar{\mathcal{L}}_{rec}^{val} = 0.02$ and for the BH2S model are $\bar{\mathcal{L}}_{rec}^{tr} = 0.005$ and $\bar{\mathcal{L}}_{rec}^{val} = 0.05$. The parameters of the network were optimized through ADAM optimizer with learning rates of the generator $\mathtt{lr}_G = 0.01$ and of the discriminator $\mathtt{lr}_D = 0.0001$. For both optimizers, a `CosineAnnealingLR` scheduler was employed in order to adjust the learning rate at training-time and improve the convergence.

The stability of the method was checked by comparing the loss curves for different training regions, we report in Fig. 5 the results obtained by varying the width of the training window in the parameter space for the XXZ and the BH model. The loss computed from the GAN reconstruction is also compared with the Kullback-Leibler divergence (KL) defined as follows:

$$D_{KL}(P||Q) = \sum_i P(x_i) \log\left(\frac{P(x_i)}{Q(x_i)}\right), \qquad (10)$$

which is a measure of the relative entropy between two probability distributions and therefore is a proper measure for the distance between two ES. The reference distribution was taken to be the ES at the origin of the phase diagram and the distance from it was computed for every ES labelled with the control variable. It is represented in Fig. 5 as a black dashed line whose values run over the rightmost y-axis. It clearly appears that for the XXZ case the KL has a delayed and slow change after the transition while for the BH model the change occurs in the right region but certainly less sharp than the loss performed with the GAN.

## Acknowledgements

D.C. gratefully acknowledges Enrico Fini for the very insightful discussions and valuable suggestions.

**Funding information**    We acknowledge support from the Deutsche Forschungsgemeinschaft (DFG), project grant 277101999, within the CRC network TR 183 (subproject B01), the European Union (PASQuanS, Grant No. 817482), the Alexander von Humboldt Foundation, from Provincia Autonoma di Trento, from Q@TN (the joint lab between University of Trento, FBK-Fondazione Bruno Kessler, INFN-National Institute for Nuclear Physics and CNR-National Research Council) and from the Italian MIUR under the PRIN2017 project CEnTraL. The MPS simulations were run on the JURECA Cluster at the Forschungszentrum Jülich, with a code based on a flexible Abelian Symmetric Tensor Networks Library, developed in collaboration with the group of S. Montangero (Padua).

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
