# Peer review of "Detection of Berezinskii-Kosterlitz-Thouless transition via Generative Adversarial Networks"

_SciPost Physics, doi:SciPost Phys. 12, 107 (2022)_

## Round 2 · Referee Report · Anonymous (Referee 1) · 2021-11-29

Strengths

he paper titiled with "Detection of Berezinskii-Kosterlitz-Thouless transition via Generative Adversarial Networks" by the authors D. Contessi, E. Ricci, A. Recati and M. Rizzi investigates the interesting topological phase transition, the so-called Berezinskii-Kosterlitz-Thouless transition, by using generative adversarial networks.

This is a research area at the intersection of physics research and machine learning artificial intelligence. On the physical side, they use spin chains, the Bose Hubbard and two-component Bose Hubbard models, which can describe cold atomic systems or other quantum many-body systems. These physical models have very rich physics and deserve to be explored.

For machine learning, the authors use the very interesting but harder to train generative adversarial network. This is a network in which the authors merged a autoencoder network and a discriminative network, and it is the first time that people use it in the field of physics research.

Report

In short, the paper also needs to be read thoroughly to make the references adequate, while making the notation more consistent throughout. This paper is very innovative and a good attempt, and it is recommended that the paper be revised and published.

Requested changes

1)The authors cite many important references, but the first paragraph does not have a single reference, leaving the reader with the feeling that many sentences are unsourced. Therefore, I suggest moving some references from the second paragraph to the first sentence of the first paragraph. This revision may make the article read more coherently.

2)In many places in the paper, the full name (abbreviation) appears many times. For example, the Generative Adversarial Network (GAN) , Autoencoder (AE) appear many times. In fact, it is sufficient for the full name to appear once in the body of the text. In the subsequent text, it is sufficient to use the abbreviation directly.

please check page 4: “auto encoder (AE)" page 5:“Figure 1: Sketch of the GAN architecture we used for the phase transition detec-tion. The Autoencoder (AE)” page 2: Our method is based on Generative Adversarial Networks (GANs) a page 2: “We develop a Generative Adversar-ial Network (GAN) architecture ” page 6: "The composite structure is a Generative Adversarial Network (GAN) [19]. and is depicted in Fig. 1."

3) "The composite structure is a Generative Adversarial Network (GAN) [19]. and is depicted in Fig. 1." Also this sentence is not written well as there is even a full stop in the middle of the sentence.

4) In page 4, they use pi to denotes reduced density matrix eigenvalue. However, in page 9, they use lambda_i. Please make these symbols consistent also check other symbols.

5)In page 6, there is a sentence " Moreover, a skip connection between two latent layers is addedto improve the model performance [22]"

Although some of the subsequent sentences describe how the skip connection is implemented, it is best to briefly explain in a few sentences why skip connection techniques can optimize the performance of neural networks. For example, this technique has been introduced as solving Residual Networks. Suppose that the left layer neurons connected by red arrows have state x, which is transformed to F(x) when they reach the right layer. If F(x) becomes zero, σ(F(x)) = σ(0) is equivalent to a constant transformation. Adding x to the right layer, σ(F(x)) becomes σ(F(x)+x) does not make the network worse, but still contains useful information in x.

6) Autoencoder is a very well known generative network, which has been used in many cases in the study of physical phase transitions. For example, the various values of the two neurons in the z-layer are used as horizontal and vertical coordinates (x_i,y_i) to draw a scatter plot and thus classify the samples (phases) in an unsupervised way. It would be nice to cite a few typical or relevant papers.

7) Still in page 4, " Their distribution can be interpretedas a Boltzmann distribution of an effective entanglement Hamiltonian, in other words the transformed eigenvalues ξi=−log(pi) constitute the energy spectrum of the entanglement Hamiltonian that takes the name of entanglement spectrum (ES). "

Please cite one or two reference papers to show how to use this formula ξi=−log(pi). [1] J.-Z. Zhao, S.-J Hu, and P. Zhang, Phys. Rev. Lett. 115, 195302 (2015). [2] T. Yoshida, R. Peters, S. Fujimoto, and N. Kawakami, Phys.Rev. Lett. 112, 196404 (2014). [3] A. M. Turner, F. Pollmann, and E. Berg, Phys. Rev. B 83,075102 (2011). [3] F. Pollmann, A. M. Turner, E. Berg, and M. Oshikawa,Phys. Rev. B 81, 064439 (2010). [4] L. Fidkowski, Phys. Rev. Lett. 104, 130502 (2010). [5] S. T. Flammia, A. Hamma, T. L. Hughes, and X.-G. Wen,Phys. Rev. Lett. 103, 261601 (2009). [6] H. Li and F. D. M. Haldane, Phys. Rev. Lett. 101, 010504 (2008)

8) In Fig.2 (a) and (b), the shapes of these entanglement spectra look very typical and beautiful, can the authors fit them quantitatively to give the formula and fitting error? It would be nice if the authors discuss the connection between the fitting formula and the CFT theory. Similar with Fig 6 and Eq 14 in this paper: https://journals.aps.org/prb/pdf/10.1103/PhysRevB.86.224406

  • validity: high
  • significance: top
  • originality: top
  • clarity: high
  • formatting: good
  • grammar: perfect

Author:  Daniele Contessi  on 2022-02-04  [id 2157]

(in reply to Report 1 on 2021-11-29)
Category:
answer to question

We acknowledge the Referee for the important remarks on the references and the notation.
We have addressed all the points in the attached "Report1.pdf".

Attachment:

Report1.pdf

---

## Round 2 · Referee Report · Anonymous (Referee 2) · 2021-12-29

Strengths

In this paper, the authors seek to detect the Berezinskii-Kosterlitz-Thouless (BKT) in several 1D quantum models using machine learning tools - specifically an autoencoder neural network. The entanglement spectrum of these quantum models is calculated in a parameter regime away from the expected BKT transition (on the Luttinger liquid side), and this serves as the training set for the autoencoder (AE). The AE learns features about the entanglement spectrum deep within a phase by reconstructing the input spectrum, as best as possible, after mapping the input through an intermediate lower-dimensional latent space ($z$ in the paper). A Generative adversarial network (GAN) is used to accelerate the training process but this does not play a role in the predictive / validation stages of this work. The authors make the last point clear in Sec.3, below eq.4. Post-training, the loss function (i.e., error) in the reconstruction process ($\mathcal{L}_{rec}$ in the paper) is claimed to serve as an effective "order parameter" for detecting the BKT transition. In other words, the transition appears as an anomaly for the trained AE where $\mathcal{L}_{rec}$ starts to increase.

  1. The approach taken by the authors is quite general as opposed to more specialized neural network (NN) architectures used in contemporary works like ref.16 that studies BKT transition in classical $q$-state clock models.

  2. The choice of using the ES as the training set, although not new (arxiv:2106.13485), is perceptive since it allows for the convenient encoding of the ground state (GS) of almost any quantum Hamiltonian. The use of ES will also naturally aid in capturing the relevant features of the GS in the vicinity of a 2nd-order quantum critical point.

  3. Furthermore, this work seeks to develop an unsupervised method to detect \emph{BKT transitions} in quantum systems, which distinguishes it from other related works in this area that primarily use supervised-learning approaches.

Weaknesses

  1. Although the main idea of the paper is to detect the BKT transition, it is not clear the method allows one to distinguish BKT from non-BKT transitions. For instance, if one used a transverse field Ising model and fed its entanglement data, would this network fail to have any anomalies or show that it is not a BKT transition? If one fed data from the disordered side (as opposed to power law correlated side), would the network fail to detect any anomalies in the loss? If one feeds data from both sides, does the network distinguish these regions?

  2. If the network is not distinguishing BKT from non-BKT transition, then it would alternatively be helpful if this approach could detect (or has the promise to detect) the BKT transition point more accurately. This has not been really shown or claimed in this paper. {For instance, the "level spectroscopy" method (arXiv: 2105.11460) which relies on the form of spectra of underlying 1+1D CFTs has been shown to very accurately capture the precise location of the BKT transition point. It is unclear if the AE network can be used in such a way.}

Report

The BKT transition has no order parameter, and its locating its position accurately usually involves high precision numerics on large system sizes, often computing quantities such as the finite size scaling of the Bose momentum distribution (XY order parameter for spins) or the superfluid stiffness with length L. Knowing whether or not a transition is of the BKT type, without making specific reference to specific operators for finite size scaling, is also interesting. This paper uses an autoencoder neural network which is trained (unsupervised) on data sets from the power-law side of the BKT transition, with a generative adversarial network (GAN) to speed up the training. The trained network is then used to show that the loss function detects the position of the transition. It is not clear to us why this method has anything to do specifically BKT physics, or what exactly is being learned say from examining the lower dimensional latent space of the AE. However this is an interesting work which may set the stage for further developments.

Requested changes

Below are some questions/ points that will benefit from clarification and, if addressed, can improve the overall quality of the paper:

  1. In figure 4, showing the performance of the AE as an anomaly detector, it seems that the reconstruction loss ($\mathcal{L}_{rec}$) is starting to increase substantially even within the validation region. However, the validation region is far from the actual transition point, which naturally raises the question of whether the AE is learning the desired features? Is the AE simply learning some irrelevant microscopic features of the GS, perhaps through overfitting and drawing conclusions based on the loss of these features? This could explain why the loss increases in the validation region and why the AE fails to detect the transition accurately. How do we ensure that the AE will not produce any false positives?

  2. Can a control be provided? A possible strategy to answer point 1 may be to validate the trained AE in a second region as control, which is deeper in the phase, and see if the increase in loss is substantially less than those presently reported. E.g., in fig.4 (b), the training can be done from ES sampled for $1<U/J<2$ (instead of $0<U<2.5$) and validated for $0<U/J<1$ as well as $2<U/J<3$. If the above point can be clarified from the results in fig.5, the authors can elaborate on it in the manuscript.

  3. What happens if the AE is trained from samples taken from the opposite regime of parameter space, e.g., in fig.4(a) if the transition is approached from the left side (lower values of $\Delta/J$)? Do the authors get a similar loss curve as the ones currently reported? If yes, can these two curves be combined to pinpoint the transition better?

  4. What is going on in the latent space? Although the authors mention in the conclusion that analysis of the features recognized by the AE will be investigated in future work, it would be nice if they could provide some insight into the latent space $z$. Given that one of the most appealing features of auto-encoders, over other NN architectures, is their ability to simplify data representation by mapping it to a lower-dimensional latent space, interpreting the features in the latent space is likely to be more feasible. It would be interesting to see if the AE recovers some physical quantities in the latent space like in arxiv 2106.13485, where the authors claim to capture the central charge from the entanglement spectrum. Perhaps, understanding the latent space could also help detect the transition point more accurately, e.g., in ref.16, where the authors can accurately detect the transition temperature by even a visual inspection of NN weights.

  5. Finally, it would be really helpful to know how this GAN/AE can be improved in future work to more accurately pinpoint the BKT transition point (e.g., such as in 2105.11460). Or are new ideas needed to make this connection?

  • validity: ok
  • significance: ok
  • originality: ok
  • clarity: good
  • formatting: good
  • grammar: good

Author:  Daniele Contessi  on 2022-02-04  [id 2158]

(in reply to Report 2 on 2021-12-29)
Category:
answer to question
reply to objection

We acknowledge the Referee for the interesting remarks and appropriate comments.
In the attached "Report2.pdf" file, we have addressed the statements that were mentioned among the weaknesses of our work and tried to make the message we wanted to convey more transparent.

Attachment:

Report2.pdf

---

## Round 3 · Author Response

We thank the Referees for the remarks.
We directly replied point by point to their comments.

---

## Round 3 · List of Changes

The list of all the changes is included in the Author replies to the Referees' reports (see files "Report1.pdf" and "Report2.pdf").

---

## Editorial Decision

published